# Improved Machine Learning Approach for Wavefront Sensing

**DOI:** 10.3390/s19163533

**Published:** 2019-08-13

**Authors:** Hongyang Guo, Yangjie Xu, Qing Li, Shengping Du, Dong He, Qiang Wang, Yongmei Huang

**Affiliations:** 1The Institute of Optics and Electronics, Chinese Academy of Sciences, Chengdu 610209, China; 2Key Laboratory of Optical Engineering, Chinese Academy of Sciences, No.1 Guangdian Road, Chengdu 610209, China; 3University of Chinese Academy of Sciences, Beijing 100049, China; 4School of Optoelectronic Science and Engineering, University of Electronic Science and Technology of China, No. 4 Section 2 North Jianshe Road, Chengdu 610054, China

**Keywords:** adaptive optics, machine learning, convolutional neural network, deconvolution

## Abstract

In the adaptive optics (AO) system, to improve the effectiveness and accuracy of wavefront sensing-less technology, a phase-based sensing approach using machine learning is proposed. In contrast to the traditional gradient-based optimization methods, the model we designed is based on an improved convolutional neural network. Specifically, the deconvolution layer, which reconstructs unknown input by measuring output, is introduced to represent the phase maps of the point spread functions at the in focus and defocus planes. The improved convolutional neural network is utilized to establish the nonlinear mapping between the input point spread functions and the corresponding phase maps of the optical system. Once well trained, the model can directly output the aberration map of the optical system with good precision. Adequate simulations and experiments are introduced to demonstrate the accuracy and real-time performance of the proposed method. The simulations show that even when atmospheric conditions D/r_0_ = 20, the detection root-mean-square of wavefront error of the proposed method is 0.1307 λ, which has a better accuracy than existing neural networks. When D/r_0_ = 15 and 10, the root-mean-square error is respectively 0.0909 λ and 0.0718 λ. It has certain applicative value in the case of medium and weak turbulence. The root-mean-square error of experiment results with D/r_0_ = 20 is 0.1304 λ, proving the correctness of simulations. Moreover, this method only needs 12 ms to accomplish the calculation and it has broad prospects for real-time wavefront sensing.

## 1. Introduction

The adaptive optics (AO) system is a technique for detecting and correcting the real-time wavefront aberration, and plays a key role in the fields of human eye vision improvement, high-speed laser transmission, and astronomical observation [1,2,3]. Phase retrieval wavefront sensing, a wavefront sensing-less (WFS-less) technology, optimizes the communication performance—without wavefront sensor. It represents one class of image-based method that restores the wavefront of an optical system when only intensity in the image plane is available [4,5,6]. Compared to the wavefront sensor (Hartmann sensor or shearing interferometry) [7,8,9], there are several advantages such as no additional optical components, low requirement for optical energy, and no special need for calibration [10]. This method has good prospects in the AO system to detect the wavefront aberrations of astronomical telescopes and free space optical communication.

The conventional image-based wavefront sensing approach is mainly classified into two general categories: Model-free and model-based technology. The former uses the control signal as the optimization variable and light intensity as a performance evaluation function. The wavefront correction is achieved by “blind” optimization of the evaluation [11]. The latter is a function-based deterministic algorithm. It attempts to find a function as the model of the WFS-less system to describe the deterministic relationship between the input and output [12]. Both of these methods are time consuming. In addition, the result depends on the original values in the iterative or iterative optimization process, which cause stagnation problem, and reduce the stability of the system.

Neural network, an image-based wavefront sensing approach, is a nonlinear, adaptive information processing system. It is composed of a large number of processing units interconnected to learn and store nonlinear mapping relationships between input and output models [13]. Initially, neural network was applied to measure optical phase distortion caused by air turbulence in AO [14], and then it was utilized to wavefront reconstruction of the Hubble Space Telescope [15,16,17]. The main contradiction of this neural network is between too much input and only one hidden layer causing poor generalization. Moreover, the lack of network layers possibly leads to over-fitting and local optimal solutions, which affect the training effect. To improve accuracy, orthogonal Tchebichef moments are introduced in discrete regions of the image coordinate space, acting as a geometric feature to process the input point spread function (PSF) directly [18]. Although the method has been proved to improve the efficiency and robustness of phase retrieval wavefront sensing, the image moment calculation increases operation time and reduces system real-time performance. Another type of model, called convolutional neural network (CNN), which is a type of deep feedforward artificial neural network, has been introduced [19]. It performs machine learning tasks with images using convolution kernels and downsampling methods. However, the real-time accuracy of CNN is still not very high. Reference [20] indicated that the root-mean-square (RMS) of wavefront error (WFE) synthesized between the real wavefront and the Zernike coefficient of the CNN prediction is about 0.2λ. The method only uses the focus image with less information. In the case of high-order coefficients fitting, there are multiple fitting results reducing the accuracy of the network. In addition, no experiments were conducted to demonstrate the feasibility of them in real situations. Nishizaki et al. proposed a generalized WFS framework that expands the design space for wavefront sensing. Using the preconditioner directly estimates Zernike coefficients of an incoming wavefront from a single intensity image [21].

However, the models above all generate the Zernike polynomial by the PSF and use the fitting Zernike coefficient error as backpropagation information. The fitting error is the wave phase difference in the whole Zernike order range. The truncation of the Zernike series may cause inaccurate prediction. On the other hand, using Zernike to characterize PSFs is an approximate fitting method essentially, where existing fitting errors will further affect the prediction accuracy. Taking these two aspects into account, we can deduce that it is hard to use this kind of machine learning model to establish the precise nonlinear mapping between the inputs and outputs.

To improve the effectiveness and accuracy of phase-based wavefront sensing, a phase -map-output CNN model is proposed. Compared with the traditional Zernike-coefficient-output neural network, there are some advantages. Firstly, as the one-to-one correspondence between input PSFs and output phase map, learning and fitting accuracy of phase-map-output CNN is higher. Secondly, the model outputs the wavefront phase map directly without additional Zernike coefficient fitting calculations, which improves the response speed of the wavefront sensing system.

## 2. The Phase Retrieval Approach Using Phase-Map-Output CNN Model

Deep neural networks are one class of the most widely applied machine learning tools, which are generally composed of a convolution part and fully connected part [22,23]. The convolution part generally contains convolution, pooling, dropout, etc. The full connection part usually contains 2 to 3 full connections at most. Finally, the classification result is obtained by softmax function, a generalization of logistic function. It maps a K-dimensional vector z of arbitrary real values to a K-dimensional vector σ(z) of real values in the range (0, 1) that add up to 1. Due to the large number of parameters in the full connection layer, it is preferred to use as little or no full connection layer as possible. The trend in neural networks is using smaller filters; deeper network depth, and little fully connected layers.

### 2.1. The Principle of the Phase-Map-Output CNN Model

In the selection of neural networks, the visual geometry group network (VGG) [24], a classic CNN, has the advantages of simple structure, smaller filters, and deeper network depth. It consists of five convolution layers, three fully-connected layers, and a softmax output layer. The layers are separated by max-pooling. The activation units of all hidden layers use the rectified linear unit (ReLU) function. The parameter calculation of VGG is reduced for fast response speed. The wider deep feature maps and more convolution kernels improve the fitting accuracy.

In order to output the aberration phase maps directly, we introduced the deconvolution layer on the basis of VGG. Deconvolution is a process to reconstruct an unknown input by measuring the output. The improved model is named the deconvolution VGG network (De-VGG). As shown in Figure 1, the De-VGG includes convolution layers, batch normalization layer filters, activation function ReLU, and deconvolution layers. The fully connected layers are removed, as the high output order slows down the calculation speed, but the deconvolution layer will not. The De-VGG outputs the aberration phase maps of the optical system with a higher real-time performance and good precision.

The structure of the De-VGG is explained as follows. The De-VGG uses 3 × 3 small convolutions kernel in all layers, whose step size is set to one. In order to obtain accurate features, the value of convolutional layer will be updated during the backward propagation process. The convolutional layer filter can be written as:
(1)$$a_{i,j}=f\left(\sum_{m=1}^{q}\sum_{n=1}^{p}\omega_{m,n}x_{i+m,j+n}+\omega_{b}\right)$$
where $a_{i,j}$ is the element of the feature map. *q* and *p* are the rows and columns of filter, which are always the same. $x_{i,j}$ is the element of row i and column j in the image. $\omega_{m.n}$ is the element of row m and column n in the weights of filter. The bias of filter is $\omega_{b}$. f represents the activation function, ReLU, which is expressed as f (x)=max(0,x). It enables the network to adapt to nonlinear functions.

The introduced batch normalized layer performs effectively to prevent the appearance of vanishing gradient problems, speeds up training, improves accuracy, and reduces overfitting. The function of the batch normalized layer is shown in the following:
(2)$${\hat{x}}_{i}=\frac{x_{i}-E\left(x_{i}\right)}{\sqrt{Var\left(x_{i}\right)}}$$
(3)$$y_{i}=\varepsilon {\hat{x}}_{i}+\sigma$$


In the function, ${\hat{x}}_{i}$ is the normalization of $x_{i}$, and $x_{i}$ is the input of the mini-batch. $E\left(x_{i}\right)=\frac{1}{m}\sum_{i=1}^{m}x_{i}$ is the mean of the mini-batch. $Var\left(x_{i}\right)=\frac{1}{m}\sum_{i=1}^{m}{\left(x_{i}-E\left(x_{i}\right)\right)}^{2}$ is the variance of the mini-batch. $y_{i}$ expresses the output of the batch normalization layer. The mean and variance of the all elements of the feature map are ε and σ. Those make the network recover the distribution of features that the original network has learned.

The 2 × 2 maximum pooling layer compresses each submatrix of the input tensor and filters out some unimportant features, reducing the calculation of the next layer and prevent overfitting. It can be expressed as:
(4)$$a_{i,j}=\max\left(x_{i,j},x_{i+1,j},x_{i,j+1},x_{i+1,j+1},\cdots \right)$$
where $x_{i,j}$ is the element of row i and column j in the image. $a_{i,j}$ is the element of the feature map.

In this model, we have designed three deconvolution layers, which transform the extracted features into the phase map. The spatial resolution of the phase map is adjusted by selecting the step size of the deconvolution layers. The last convolutional layer works as the output layer. This structure implements the input of the PSFs and the output of the phase maps.

### 2.2. Generation of Data Set

To validate the feasibility of the wavefront sensing method based on the De-VGG network, the data set for training is needed to establish the mapping relation between the intensity images and coordinates of the image plane. In the OA system, the image of the focal plane based on the image-forming principle can be written as the following:
(5)$$u\left(x,y\right)={\left|o\left(x,y\right)⊗h\left(x,y\right)\right|}^{2}$$
where u(*x*, *y*) is the image of the focal plane, o(*x*, *y*) is the ideal field distribution of the objective, ⊗ is a convolution operation, and (*x*, *y*) represent the coordinates of the image plane. h(*x*, *y*) is the pulse response and it can be expressed as:
(6)h(x,y)=ℱ{P(x,y)φ(x,y)}
where ℱ is the Fourier transform, P(*x*,*y*) is the transmittance function of pupil, and is the modulated phase. Based on the Fourier optics imaging principle, the PSF can be expressed as:
(7)$$\mathrm{PSF}\left(x,y\right)={\left|h\left(x,y\right)\right|}^{2}$$
when the image is on the focal plane, the defocus plane image can be equivalent to add an additional phase to the wavefront, the additional phase Δφ can be written as:
(8)$$\Delta \varphi =-\frac{k*d}{2*\left(f+d\right)*f}*\left(u^{2}+v^{2}\right)$$
where d is focal shift, f is the focus length, u and v is the coordinates of the pupil plane. Based on the analysis above, we can set up the data set. In order to guarantee the uniqueness of the wavefront phase solution, a pair of PSF images obtained on different focal planes is required.

### 2.3. The Working Procedure of the De-VGG

The working procedure of the De-VGG wavefront sensing is illustrated in Figure 2. Firstly, we determine the system parameters, including wavelength, aperture size, focal length, pixel of detector, and defocus length. Secondly, we generate the training set. The input of the De-VGG is a set of 224 × 244 PSF images. All PSFs are normalized in the range of 0~1 before entering into the De-VGG. Depending on the back-propagated gradients, the learned parameters of the De-VGG are updated from a mini-batch of inputs. Thirdly, the number of neurons in each layer is selected. The neural network is trained with the input data sets and the corresponding output data sets. Machine learning relies on a stochastic gradient descent algorithm, and the learning rate parameter controls the step size of the update. Adam, a gradient-based stochastic optimization algorithm, is used to minimize the learning rate which is 0.0001 and adaptively updates according to the value of the gradient. The minimized loss function of the designed De-VGG is the difference of pixel mean square (PMS) between the predicted and true phase map. After updating the parameters, the average loss of the PSFs is determined, called the verification phase. Compared with training losses, the inspection verification loss indicates to us whether the De-VGG is over-fitting. Finally, the well-trained neural networks are applied to determine the phase maps from the PSF images, which is collected from the optical system under different atmospheric turbulence conditions.

## 3. Simulation

In the AO system, the influence of the turbulence can be described by atmospheric coherence length r0. The atmospheric conditions are better as the value of r0 increases, which means the turbulence has less effect on the laser. When the diameter D of the receiving telescope is less than one meter, we define as follows. When D/r_0_ = 2, it means the level of turbulence is low, D/r_0_ = 10 is medium, and D/r_0_ = 20 is strong [25]. Directly estimating the wavefront comes at the expense of choosing the spatial resolution of the phase map. When we calculate the phase map, we are free to choose at what resolution to reconstruct. That resolution will fix the amount of output parameters that are estimated, making the training easier or harder, and generating some degree of over-fitting. To realize strong turbulence sensing, we set the 135 × 135 resolution of the output phase map.

To realize the wavefront aberration measurement, the De-VGG is utilized to recognize the PSF images of the focal plane and defocus plane. And the network extracts the corresponding phase map. As the global piston and tip or tilt terms can be estimated rapidly using centroiding algorithms or other registration methods [26], the 4~64th Zernike polynomial is chosen to illustrate the PSFs. Initially, the model of the PSFs is under approximately strong turbulence, whose D/r_0_ = 20, where 20,000 epochs were trained. To determine the practicality of the model, we collected 2000 pairs in focus and defocus plane as test sets separately with $D/r_{0}$ = 6, $D/r_{0}$ = 10, $D/r_{0}$ = 15, and $D/r_{0}$ = 20. The phase maps between the simulation and the reconstruction by the De-VGG in different situations are shown in Figure 3, Figure 4, Figure 5 and Figure 6. Atmospheric wavefronts for $D/r_{0}$ = 6, 10, 15, and 20 are respectively randomly generated. The initial PSF images in focus plane and defocus plane are shown in Figure 3a, Figure 4a, Figure 5a and Figure 6a and Figure 3b, Figure 4b, Figure 5b and Figure 6b, the initial wavefront maps are shown in Figure 3c, Figure 4c, Figure 5c and Figure 6c, and the predicted wavefront by the proposed De-VGG network are shown in Figure 3d, Figure 4d, Figure 5d and Figure 6d.

To verify the fitting effect of the model, in addition to RMS, we introduce the normalized-pixel-mean-square (NPMS). It is the normalized mean square of pixels between the output phase maps of the De-VGG and the corresponding input PSF images. NPMS of 2000 test sets was taken as the model evaluation index. As shown in Table 1, the NPMS is 0.0067 and the RMS is 0.1307 λ with $D/r_{0}$ = 20, which has a better accuracy than existing neural networks. When the $D/r_{0}$ = 15, the NPMS is 0.0041 and the RMS is 0.0909 λ. The RMS is 0.0718 λ with $D/r_{0}$ = 10, proving that the De-VGG can fit the phase map generated by atmospheric turbulence perfectly. The result is consistent with the actual situation and it certificates the applicability of the De-VGG network.

Furthermore, to prove the superiority of the model, we compared it with the stochastic parallel gradient descent (SPGD) algorithm. SPGD calculates the variation of image quality evaluation function, and corrects the control parameters according to the iterative formula. The computation latency of the De-VGG and traditional SPGD method is shown in Figure 7. The latency is calculated on a 108Ti Graphics Processing Unit (GPU). The sensing time of the charge-coupled device (CCD) camera (Watec WAT-902H2, Cremona, Italy) and the correction time of the corrector are not considered. To reach the same RMS, it takes SPGD 304 ms corresponding to $D/r_{0}$ = 6 with 0.0703 λ. The running time is much longer than the De-VGG because SPGD requires hundreds of iterations, while De-VGG does not. With the increase of turbulence intensity, the correction ability based on the SPGD algorithm is greatly reduced. When D/r_0_ = 10, the system can barely converge after 1000 iterations and the running time is about 448 ms; when D/r_0_ = 15, the system still cannot converge after 1000 steps of iteration. The convergence speed is further reduced, and the correction capability cannot be fully utilized. 

## 4. Experimental Validation

### 4.1. Experimental Setup

In order to better analyze and study the performance of the De-VGG phase retrieval technology, we built a measurement system of wavefront detection based on liquid crystal spatial light modulator (LCSLM). The LCSLM we used is pluto-2-NIR-011 produced by Holoeye, with 1920 × 1080 pixels and 8 μm electrode spacing. The optical structure is shown in Figure 8a. The experimental optical system is mainly composed of three parts, including the coherent point source, aberration wavefront simulator, and phase diversity wavefront sensor. The coherent point source is mainly composed of a laser, a spatial filter (SF), and a collimator (C). The pinhole of SF is placed at the focal plane of the collimator to simulate a point source at infinity. The aberration simulator is mainly used to change the wavefront distribution of the ideal beam. It consists of an aperture diaphragm (AD), a polarizer (P), a beam splitter (BS), and LCSLM. The AD is used to limit and determine the aperture shape, size, and position of the incident parallel beam. Phase diversity wavefront sensor mainly includes lenses (L), CCD, and translation guide rail. The modulated light is reflected through the lens and incident on the CCD, whose position is controlled by the guide rail. When the CCD is adjusted to the focal plane of the lens, it can acquire images of intensity distribution of the focal plane. When the CCD is adjusted to a defocus plane of known defocusing, the defocus plane images are acquired. The experimental platform of the optical path is shown in Figure 8b.

The main optical device parameters are shown in the Table 2. A fiber laser is used as the light source with 850 nm wavelength in the experiment. The image intensity distribution is captured by a CCD camera (WAT-902H2), whose pixel size is 8.6μm×8.3μm.

Before testing, the center of the focus plane and the defocus plane are calibrated. Firstly, a control image with a gray value of 0 is generated in the effective aperture range and loaded into the LCSLM. At the focal plane position of the lens, a focal plane intensity distribution image is acquired using a CCD camera. Then, the translation guide rail is adjusted, and the photosensitive surface of the CCD is moved to the position of the defocus plane. The defocusing amount introduced by the guide rail is recorded. Adjusting the intensity of the irradiation source appropriately, the image of the intensity distribution of the defocused surface is collected. Finally, the center position of the spot in the focal plane and the defocus plane is calculated, and the light intensity image captured by the CCD camera is correspondingly translated and cropped according to the center position.

### 4.2. Application of the De-VGG and Result Analysis

Finally, the obtained De-VGG neural network applies to the real PSF images collected from the experimental optical system. The De-VGG outputs the aberration phase maps of the optical system. In the experiment, 22000 pairs of PSFs in focal plane and defocus plane are respectively collected in the case of $D/r_{0}$ = 20. The 20000 pairs of PSFs are the training set and the other 2000 pairs of PSFs are the testing set. The effectiveness of the proposed approach is validated using the following two methods.

On the one hand, the recovered aberration phase maps are used to re-generate a pair of PSF images according to Fourier optics. Then we load the reconstructed phase maps on LCSLM, and use the experiment platform to obtain the real PSF images. The phase maps of the initial and the reconstruction maps of the De-VGG are shown in Figure 9. The effectiveness of the proposed approach can be qualitatively validated by comparing this pair of generated PSF images with those real PSF images collected from the optical system. In this experiment, the comparison is shown in Figure 10. The initial PSF images in focus plane and defocus plane are shown in Figure 10a,c respectively, the in focus PSF images by De-VGG predicted are shown in Figure 10b, and the predicted defocus PSF images are shown in Figure 10d. It can be recognized that the collected PSF images bear strong similarities with those regenerated PSF images, which can qualitatively validate the accuracy and effectiveness of the proposed approach.

On the other hand, the model indicators including NPMS, RMS, and running time are listed in Table 3. The average value of NPMS for the 2000 sets of the experiment is 0.0066 and RMS is 0.1304 λ, which can definitely demonstrate the effectiveness and accuracy of the proposed approach. The running time of the De-CNN networks in each testing image is approximately 12 ms, that demonstrate the real-time performance.

## 5. Conclusions

A novel phase retrieval neural network named the De-VGG is proposed in the area of wavefront sensing. The model transforms the coefficient approximation into a phase map calculation. Once well trained, the De-VGG neural network can output the aberration phase map of the optical system directly. Compared to the conventional iterative phase retrieval approaches, this method is higher in efficiency and calculating speed. Compared to the current image-based phase retrieval approaches which are trained to predict Zernike coefficients using machine learning, the De-VGG is more precise and more practical. Simulation and experiment are implemented to determine that the De-VGG can be trained to estimate wavefront distortion efficient and accuracy.

Currently, the De-VGG does not consider tip or tilt terms. Powerful learning ability can be used to continue training aberrations including tip or tilt for better applicability. Future work is considered to generate the aberration correct phase map, which is used to reconstruct the distortion spot in the AO system. For dynamic aberration correction, the PSFs captured in the existing experiments contain static aberrations that are detrimental to the dynamic estimation of the neural network. Combining the static aberrations with dynamic aberrations for phase retrieval sensing is considered.

This work presents a feasible and easy-implemented method to improve the efficiency and accuracy of the phase retrieval wavefront sensing. It contributes to the application of machine learning methods to the area of the AO system.

## Figures and Tables

**Figure 1 sensors-19-03533-f001:**
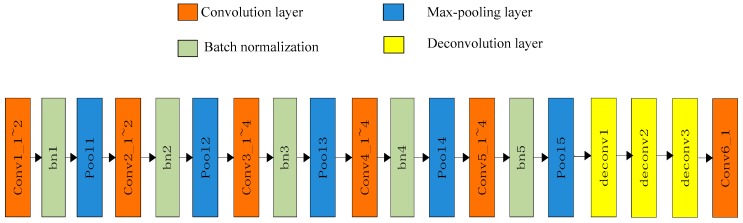
Architecture of the deconvolution VGG network (De-VGG )network.

**Figure 2 sensors-19-03533-f002:**
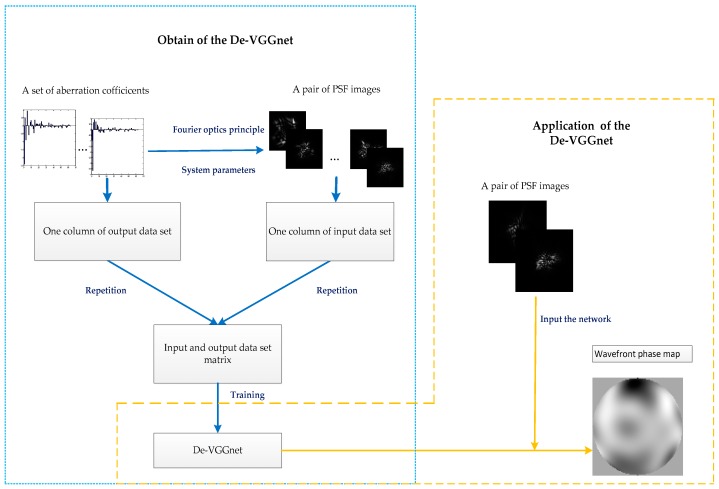
Working procedure of the De-VGG wavefront sensing approach.

**Figure 3 sensors-19-03533-f003:**
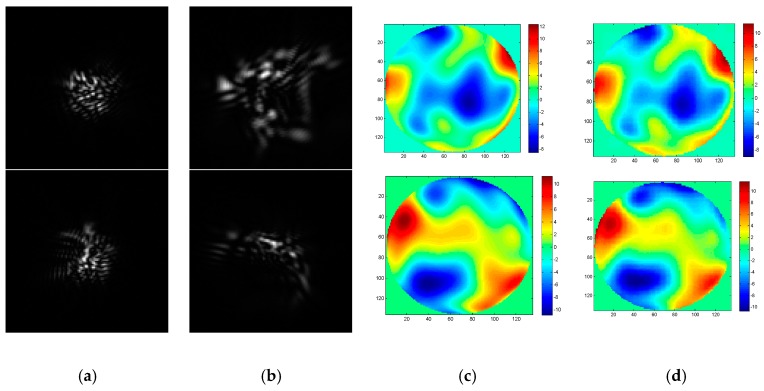
Predicting effect of (**a**) the initial in focus input point spread function (PSF) images; (**b**) the initial defocus PSF images; (**c**) the initial wavefront maps; (**d**) the wavefront maps by the De-VGG predicted for $D/r_{0}$ = 20.

**Figure 4 sensors-19-03533-f004:**
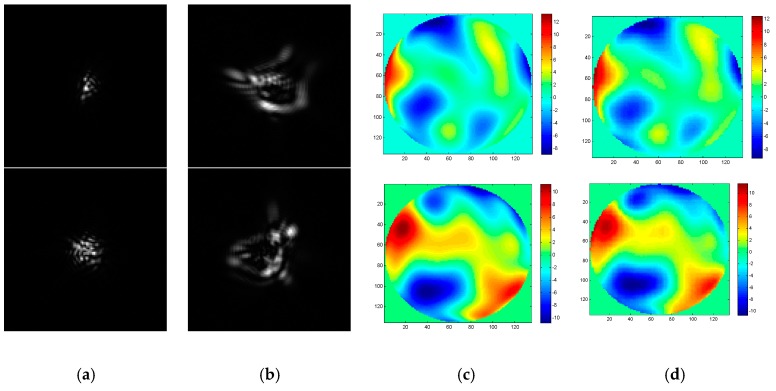
Predicting effect of (**a**) the initial in focus PSF images; (**b**) the initial defocus PSF images; (**c**) the initial wavefront maps; (**d**) the wavefront maps by the De-VGG predicted for $D/r_{0}$ = 15.

**Figure 5 sensors-19-03533-f005:**
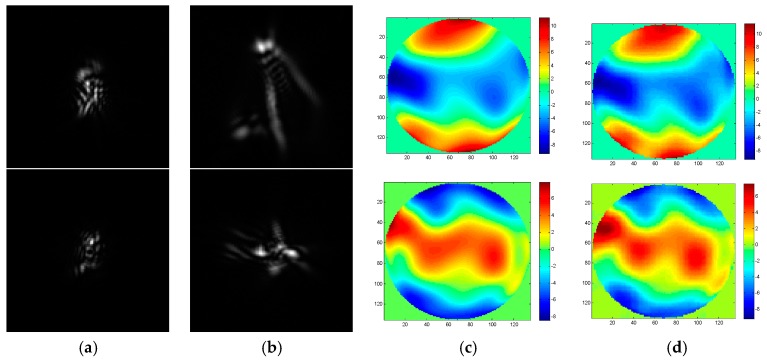
Predicting effect of (**a**) the initial in focus PSF images; (**b**) the initial defocus PSF images; (**c**) the initial wavefront maps; (**d**) the wavefront maps by the De-VGG predicted for $D/r_{0}$ = 10.

**Figure 6 sensors-19-03533-f006:**
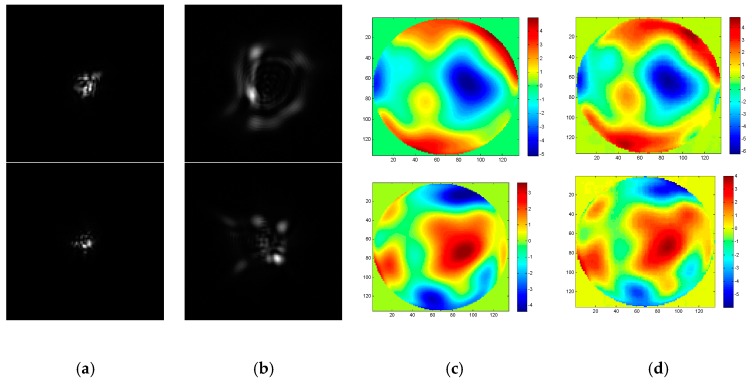
Predicting effect of (**a**) the initial in focus PSF images; (**b**) the initial defocus PSF images; (**c**) the initial wavefront maps; (**d**) the wavefront maps by the De-VGG predicted for $D/r_{0}$ = 6.

**Figure 7 sensors-19-03533-f007:**
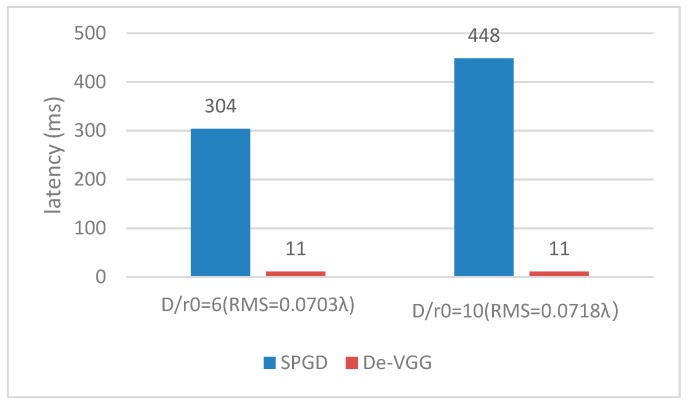
Root mean square (RMS) with latency of the stochastic parallel gradient descent (SPGD) algorithm and De-VGG.

**Figure 8 sensors-19-03533-f008:**
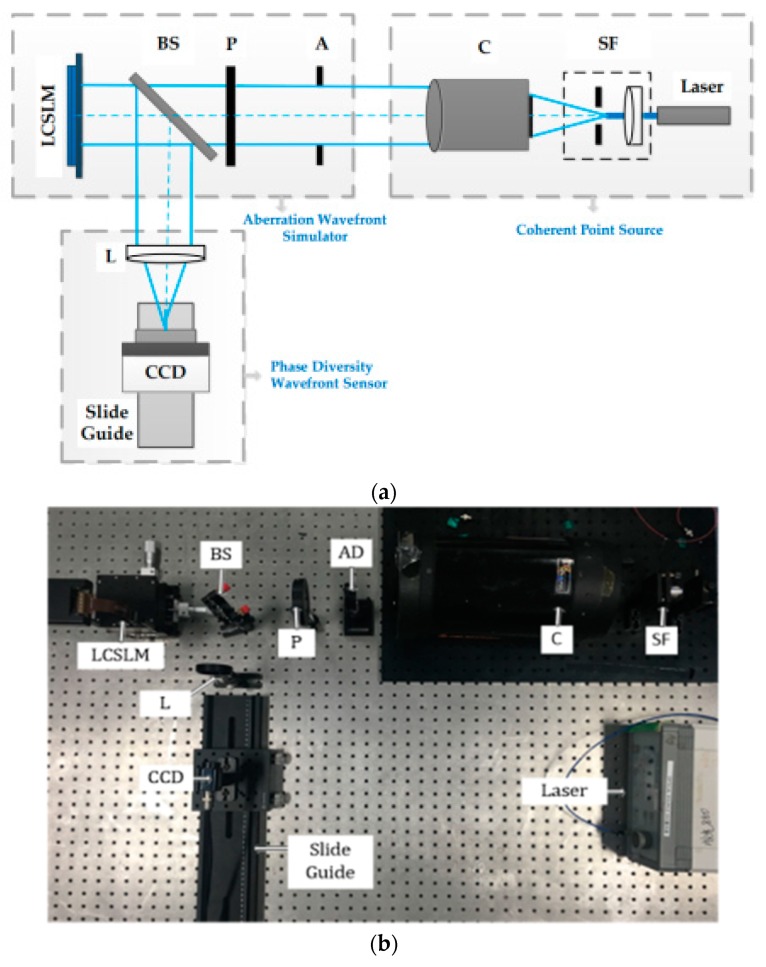
The schematic diagram (**a**) and physical map (**b**) of the optical system used in the experiment. P: Polarizer, SF: Spatial filter, C: Collimator, AD: Aperture stop, BS: 1:1 Spectroscope, CCD: Imaging detector, L: Lens.

**Figure 9 sensors-19-03533-f009:**
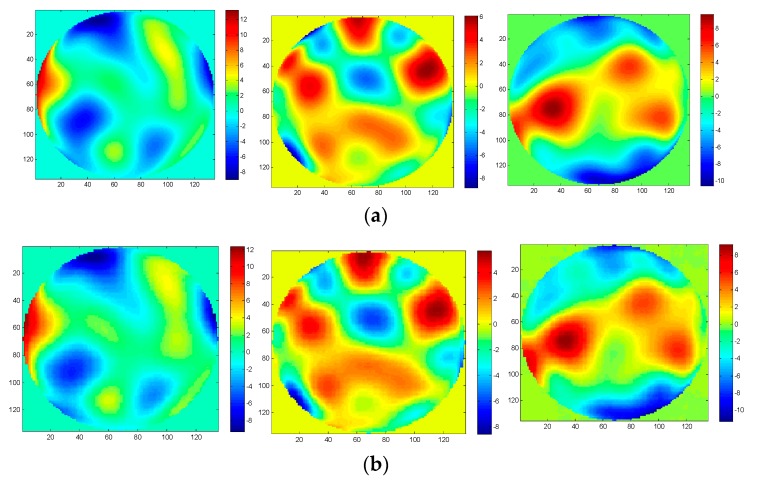
Phase maps of (**a**) the initial wavefront; (**b**) the wavefront by the De-VGG predicted for $D/r_{0}$ = 20.

**Figure 10 sensors-19-03533-f010:**
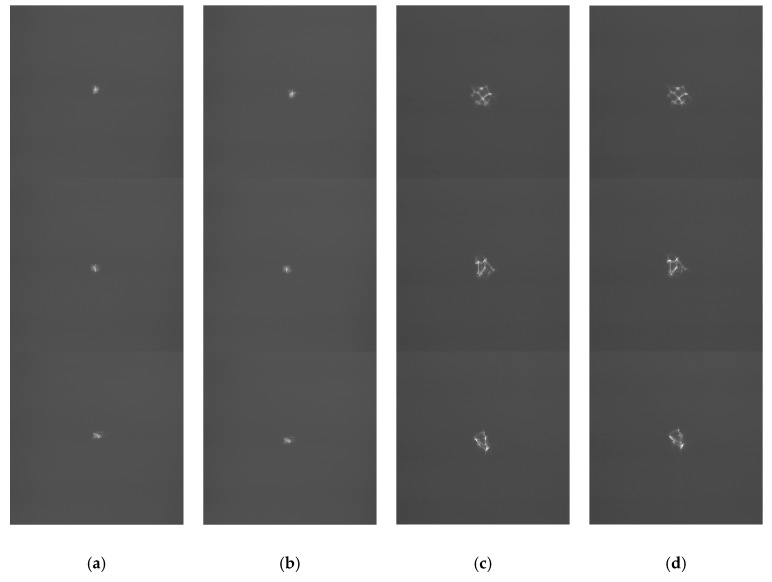
PSF images of (**a**) the initial in focus plane; (**b**) the in focus plane by the De-VGG predicted; (**c**) the initial defocus plane; (**d**) the defocus plane by the De-VGG predicted for $D/r_{0}$ = 20.

**Table 1 sensors-19-03533-t001:** Normalized-pixel-mean-square (NPMS) under different test sets of the reconstructed and simulated wavefront.

Atmospheric Conditions	NPMS (Testing Set)	RMS (Testing Set)
$D/r_{0}$ = 20	0.0067	0.1307 λ
$D/r_{0}$ = 15	0.0041	0.0909 λ
$D/r_{0}$ = 10	0.0029	0.0718 λ
$D/r_{0}$ = 6	0.0025	0.0703 λ

**Table 2 sensors-19-03533-t002:** Parameters of experimental optics.

Symbol	Name	Focal Length	Focal Length
L	lens	200 mm	50 mm
A	aperture diaphragm	N/A	36 mm
BS	50/50 beam splitter	N/A	50 mm

**Table 3 sensors-19-03533-t003:** The model indicators of the experiment ($D/r_{0}$ = 6).

Atmospheric Condition	NPMS (Testing Set)	RMS (Testing Set)	Running Time
$D/r_{0}$ = 20	0.0066	0.1304 λ	~12 ms

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
