# Peer review of "Improved Machine Learning Approach for Wavefront Sensing"

_sensors, 2019, doi:10.3390/s19163533_

Round 1

Reviewer 1 Report

This is an interesting manuscript on the use of machine learning for wavefront sensing. The paper combines modelling with experimental proof-of-principle verification and is of interest to the readers of Sensors. The manuscript needs significant proof reading, preferably by a native speaker as there are many English writing errors in nearly every sentence. However, the study is for the most clear enough to be understood. I would encourage the authors to get professional help with language editing. 

My other comments are as follows:

1) In the introduction, the authors are being very sceptical about the Hartmann-Shack wavefront sensor. Yet it is widely used which to itself proves that it often works well. Maybe reduce the negative comments on the HS-WFS performance as clever methods can be used for centroiding and to supress noise. 

2) Blind AO algorithms work well also with confocal detection, which is not mentioned. For example, see Jewel et al. "A direct comparison between a MEMS deformable mirror and a liquid crystal spatial light modulator in signal-based wavefront sensing" Journal of the European Optical Society 8 (2013).

3) FIenup's group in Rochester has pioneered a number of machine learning - wavefront sensing studies. It is not made clear enough how the method presented here differs from studies done in the Rochester group such as Paine and Fienup (Ref. 20) or relatedly Nishizaki et al "Deep learning wavefront sensing" Opt. Express 27 (2019).

4) It would be very useful if the authors would cite a reference on the VGG method described in section 2.1. At the moment this section is not easy to read, without searching for more details elsewhere. Clarify should be improved. What is new here by the authors, and what has been presented by others? Please add reference(s).

5) Relatedly the Softmax function should be described in the beginning of Sec. 2. This is not common knowledge.

6) In the introduction, it is stated that different combinations of Zernike coefficients can give the same wavefront. This is not strictly true, as the Zernike polynomials are orthogonal functions. Consider removing, or explain that this is due to the truncation of the Zernike series.

7) In section 2.2 it is stated that the number of neurons is corrected selected. How can you make sure that you have chosen the correct number? Please clarify.

8) In the simulation section it refers to "intuitively" shown in Figs. 3 - 6. Remove intuitively.

9) In the experimental validation section, please describe which LCSLM you use. Also, the light is from a laser, and thus likely to be coherent. Why do you write incoherent point source? This is wrong.

10) Please correct the system figure, to show a properly shaped lens to collimate the light after the spatial filter. 

11) How do you remove, or deal with, diffraction orders from the LCSLM?

12) If this system is to replace the Hartmann-Shack WFS please comment on how it will work with extended guide stars as often used in astronomy. 

Author Response

Thank you very much for your review. We have carefully revised and responded to your comments as follows:   In the introduction, the authors are being very sceptical about the Hartmann-Shack wavefront sensor. Yet it is widely used which to itself proves that it often works well. Maybe reduce the negative comments on the HS-WFS performance as clever methods can be used for centroiding and to supress noise. 

Response 1According to suggestion, we have reduced the negative comments on the HS-WFS performance and used a more pertinent way to express.

Blind AO algorithms work well also with confocal detection, which is not mentioned. For example, see Jewel et al. "A direct comparison between a MEMS deformable mirror and a liquid crystal spatial light modulator in signal-based wavefront sensing" Journal of the European Optical Society 8 (2013).

Response 2:In the introduction, we have added a summary of the existing wavefront algorithm to further prove the significance of the method proposed in this paper.

Blind AO algorithms work well but time consuming. In addition, the result depends on the original values in the iterative or iterative optimization process, which may cause stagnation problem, and reduce the stability of the system.

FIenup's group in Rochester has pioneered a number of machine learning - wavefront sensing studies. It is not made clear enough how the method presented here differs from studies done in the Rochester group such as Paine and Fienup (Ref. 20) or relatedly Nishizaki et al "Deep learning wavefront sensing" Opt. Express 27 (2019).

Response 3:The difference with Fineup (Ref. 20) is that he only uses the focus image. In the case of high-order coefficients fitting, the focus image provides less information and will produce a Zernike coefficient fitting error. Such fitting results will reduce the accuracy of the network.

Compared with Nishizaki, this method uses different network structures to directly output the wavefront map corresponding to the PSF instead of outputting the zernike coefficient.

It would be very useful if the authors would cite a reference on the VGG method described in section 2.1. At the moment this section is not easy to read, without searching for more details elsewhere. Clarify should be improved. What is new here by the authors, and what has been presented by others? Please add reference(s).

Response 4: Firstly, We have cited a reference on the VGG method described in section 2.1. Clarify have been improved to make the article easy to read.

We have proposed a new network structure on the basis of VVG. The latter three convolutional layers are replaced to deconvolution layers, so that the network outputs the phase map directly.

The structure of this network is simple and easy to implement, and the prediction effect is improved compared with the commonly Zernike-coefficient-output method.

5)Relatedly the Softmax function should be described in the beginning of Sec. 2. This is not common knowledge.

Response 5: We have added the description of the Softmax function in the beginning of Sec. 2.

6) In the introduction, it is stated that different combinations of Zernike coefficients can give the same wavefront. This is not strictly true, as the Zernike polynomials are orthogonal functions. Consider removing, or explain that this is due to the truncation of the Zernike series.

Response 6: According to your suggestion, we have made correction that this is due to the truncation of the Zernike series in the article.

In section 2.2 it is stated that the number of neurons is corrected selected. How can you make sure that you have chosen the correct number? Please clarify.

Response 7: For the neural network proposed in this paper, most of the former network structure comes from VGG, which has been validated by a large number of scholars. The latter deconvolution part is calculated based on the output, and there may be a better solution in this part. At present, our work is mainly to verify the feasibility of this method. The design of the optimal number of neurons will be explored in the following work.

8) In the simulation section it refers to "intuitively" shown in Figs. 3 - 6. Remove intuitively.

Response 8: Thanks for your correction, “Intuitively” is removed.

In the experimental validation section, please describe which LCSLM you use. Also, the light is from a laser, and thus likely to be coherent. Why do you write incoherent point source? This is wrong.

Response 9: The LCSLM we used is pluto-2-NIR-011 produced by Holoeye, with 1920×1080 pixels and 8electrode spacing.

Thanks for your correction, and we have corrected the type of light in the article.

Please correct the system figure, to show a properly shaped lens to collimate the light after the spatial filter. 

Response 10: We have corrected the system figure to show a properly shaped lens.

11) How do you remove, or deal with, diffraction orders from the LCSLM?

Response 11: First, we have calibrated the LCSLM to obtain the accurate relationship between the gray value and the modulation phase. Secondly, a polarizer is added to the optical path. It is used to control the polarization state of the incident light and optimize the diffraction efficiency of the LCSLM. Third, the diffraction orders from the LCSLM are considered as a part of the distortion and also within the learning range of the neural network proposed in this paper. 

12) If this system is to replace the Hartmann-Shack WFS please comment on how it will work with extended guide stars as often used in astronomy. 

Response 12: The system collects the far-field light intensity through the CCD, and inputs far-field light information into the neural network to generate a corresponding surface shape.  Finally, the system controls the liquid crystal for compensation.

In the process of observing the guide star, the CCD needs to have a higher sampling frequency to ensure that clearer guide information can be collected. Second, the neural network data processing frequency should be sufficient to ensure the bandwidth of the system. Thirdly, it is necessary to ensure the execution frequency of the liquid crystal for higher real-time correction.
In summary, the system can basically predict the PSF based on the acquired far-field information, and can quickly output the corresponding distortion phase map. It is expected to be applied to astronomical observation applications after the system is perfected and strengthened.

Reviewer 2 Report

The manuscript written by Guo et al. proposes to use machine learning to determine the wavefront in the adaptive optics system. The authors demonstrated their methods using both numerical and experimental approach and showed that the wavefront can be determined accurately. More importantly, the computational time was quantified to be only 12 milliseconds, which shows the promise for real-time applications. In general, the manuscript is well written and easy to follow. The data presented in this manuscript are also solid and support the authors’ claim. Therefore, the manuscript can be considered for publication. Some minor comments are listed below.

1.       In the introduction part, the authors wrote “The main contradiction is that the light intensity is dispersed due to excessive sensor sub-aperture, …”. First of all, I do not quite understand this statement. Moreover, there a several major challenging in the existing adaptive optics systems. Therefore, I suggest to change “The main contradiction …” to “One of the major challenging in the adaptive system ….”.

2.       By observing Eq. 5, it seems that the authors computes the convolution in an incoherent manner (intensity summation rather than field summation). If that is the case, please be more specific

3.       Color bars should be provided for the gray scale images in the manuscript, especially for the phase maps.

4.       What is the purpose of adding the images with respect to defocus here?

5.       The PSF images obtained during experiments have observable background. What are the causing factors?

6.       In the adaptive optics system, the complexity of the wavefront is relatively low. I wonder whether this machine learning based approach can be extended to the regions where strong scattering exist. See the following paper “Focusing light through biological tissue and tissue-mimicking phantoms up to 9.6 cm in thickness with digital optical phase conjugation”, Journal of Biomedical Optics, 21(8), 085001 (2016) as an example.

Author Response

Thank you very much for your review. We have carefully revised and responded to your comments as follows:

1.In the introduction part, the authors wrote “The main contradiction is that the light intensity is dispersed due to excessive sensor sub-aperture, …”. First of all, I do not quite understand this statement. Moreover, there a several major challenging in the existing adaptive optics systems. Therefore, I suggest to change “The main contradiction …” to “One of the major challenging in the adaptive system ….”.

Response1:According to your suggestion, we have reduced the negative comments on the HS-WFS performance and used a more pertinent way to express. And in the introduction, we have added a summary of the existing wavefront algorithm to further prove the significance of the method proposed in this paper.

“The main contradiction is that the light intensity is dispersed due to excessive sensor sub-aperture, …”is explained as follows:During the Hartmann detection process, the incident wavefront reaches the microlens array and is dispersed. The dispersed beams are focused by a microlens onto the CCD. Too many sub-apertures will excessively disperse the light intensity. When the light intensity is weak or flickering, it may cause the Hartmann detection to fail.

2.By observing Eq. 5, it seems that the authors computes the convolution in an incoherent manner (intensity summation rather than field summation). If that is the case, please be more specific

Response2:Thank you very much for your correction. In this paper, the expression of Equation 5 is improper. The system used in this paper is coherent imaging.This system is a parallel light incident, equivalent to a point source with infinity. In the case of a point source, o(x,y) is approximately equal to Pulse function.

We have modified the description of Eq. 5 and adjusted Section 2.1 to make the article more rigorous and easy to read.

3.Color bars should be provided for the gray scale images in the manuscript, especially for the phase maps.

Response3:Color bars are provided for the phase maps in the manuscript.

4.What is the purpose of adding the images with respect to defocus here?

Response 4:Defocus images are used to solve the multi-solution problem of the focus. In the case of high-order coefficients fitting, the focus image provides less information and will produce a Zernike coefficient fitting error. Such fitting results will reduce the accuracy of the network.

5.The PSF images obtained during experiments have observable background. What are the causing factors?

Response 5:The PSF images obtained during experiments in the daytime (with background light). In order to make the comparison of the experimental results clearer, we have reacquired the image without background light to enhance the readability of the results.

6.In the adaptive optics system, the complexity of the wavefront is relatively low. I wonder whether this machine learning based approach can be extended to the regions where strong scattering exist. See the following paper “Focusing light through biological tissue and tissue-mimicking phantoms up to 9.6 cm in thickness with digital optical phase conjugation”, Journal of Biomedical Optics, 21(8), 085001 (2016) as an example.

Response 6:The premises of machine learning proposed in this paper are a one-to-one or many-to-one input-output relationship, and a sufficient training set for training the network.

It can be seen from the references that this strong scattering satisfies the basic requirements of this machine learning. This method we proposed currently achieves predictions under strong atmospheric turbulence conditions and is expected to achieve strong scattering applications after improvement.

However, we also found that when the scattering intensity increases to a certain extent (the chicken thickness reaches 3.0cm in the reference literature), the background influence is greatly increased. How to remove or reduce the influence of the background in training is a key problem that needs to be solved in strong scattering applications. 

Round 2

Reviewer 1 Report

The clarity of the manuscript has improved significantly and I appreciate the careful consideration of each point raised in the review. The authors may still want to make a small graphical correction to Fig. 10 (column c) where two sub-images slightly overlap, but this is a very minor issue.